# Communicating biopsy results from breast screening assessment: current practice in English breast screening centres and staff perspectives of telephoning results

Sian Z Williamson,[1] Rebecca Johnson,[2] Harbinder K Sandhu,[3] David R Ellard  ,[3] Jacquie Jenkins,[4] Margaret Casey,[5] Olive Kearins,[6] Sian Taylor-Phillips  [7]

For numbered affiliations see end of article.

**Correspondence to**
Dr Sian Taylor-Phillips;
s.taylor-phillips@warwick.ac.uk

## ABSTRACT

**Objective** To record how breast screening centres in England deliver all biopsy results (cancer/non-cancer) from the breast assessment visit.

**Design** Online survey of 63 of 79 breast screening centres in England from all regions (East Midlands, East of England, London, North East Yorkshire & Humber, North West, South East, South West, West Midlands). The survey contained quantitative measures of frequency for telephoning biopsy results (routinely, occasionally or never) and optional qualitative free-text responses. Surveys were completed by a staff member from each centre.

**Results** There were no regional trends in the use of telephone results services, ($X^2$ (14, n=63)=11.55, p=0.64), Centres who telephoned results routinely did not deliver results sooner than centres who deliver results in-person ($X^2$ (16, n=63)=12.76, p=0.69).
When delivering cancer results, 76.2% of centres never telephone results and 23.8% of centres occasionally telephone results. No centres reported delivering cancer results routinely by telephone. Qualitative content analysis suggests that cancer results are only telephoned at the patient request and under exceptional circumstances. When delivering non-cancer results, 12.7% of centres never telephoned results, 38.1% occasionally telephoned results and 49.2% routinely telephoned results. Qualitative content analysis revealed different processes for delivering telephone results, including patient choice and scheduling an in-person results appointment for all women attending breast assessment, then ringing non-cancer results unexpectedly ahead of this prebooked appointment.

**Conclusions** In the National Health Service Breast Screening Programme, breast assessment results that are cancer are routinely delivered in-person. However, non-cancer breast assessment results are often routinely delivered by telephone, despite breast screening policy recommendations. More research is needed to understand the impact of telephoning results on women attending breast assessment, particularly women who receive a non-cancer result. Future research should also consider how women themselves might prefer to receive their results.

## Strengths and limitations of this study

► This study gives an up-to-date picture of current results-giving practice of National Health Service Breast Screening Programme centres nationally.

► A high response rate was achieved, so results are generalisable to the English National Health Service Breast Screening Programme.

► The qualitative comments made by National Health Service staff gave deeper insight into how telephone results are delivered in practice.

► Survey responses were subjective and not checked against centre policy documents.

► Formal analysis of the concordance between communication practice and policy guidelines was not conducted

## BACKGROUND

Breast cancer is one of the most common cancers internationally.[1] The National Health Service Breast Screening Programme (NHSBSP) was launched to aid the early detection of breast cancer at the population level because early detection is linked with better prognosis.[2] At the screening, a mammogram (x-ray) is performed on each breast.[3] If an abnormality is found during this screening mammogram, women will be recalled to attend an assessment for further tests. These tests can include a core needle biopsy, which involves the removal of sections of tissue from the suspicious breast region which are sent for cytological examination. A biopsy is the definitive test at breast assessment to confirm if the mammogram abnormalities found are cancer. In 2016–2017, a total of 2 199 342 women in England were screened by the NHSBSP. Of these, 89 104 women were referred for assessment and 40 255 women had a core biopsy.[4]

In recent years, the NHSBSP has been considering which communication method might be preferable for delivering non-cancer biopsy results from breast assessment. The NHSBSP service specification recommends that all breast assessment results should be delivered in-person, which includes cancer and non-cancer results.[5] Furthermore, the guidelines state that telephone results should only be used at the patients request and should not be standard practice.[6] Despite these recommendations, some centres already routinely deliver non-cancer assessment results by telephone.[5 7] There is ongoing concern about the impact of delivering a non-cancer assessment result by telephone. One of the main concerns is how anxious women feel when receiving a result, even when the result is non-cancer.[8] Another concern is the potential for miscommunication by telephone.

In other areas of healthcare, a variety of communication methods are used to deliver results, including in-person consultations, telephone, letters and email. Each method of communication used in results delivery presents different advantages, disadvantages and challenges in implementation. Results delivered in-person are often seen as the 'gold standard'.[9] However, as technology advances, fewer healthcare results are now delivered in-person. Liederman et al[10] stated that 'face-to-face contact is not necessary for effective communication' (pg. 52). Most commonly, results delivery is moving towards telephone and telemedicine.[11] Despite this, Cochrane review evidence suggests that we still do not know enough about the impact of telephone results services on healthcare outcomes.[12]

There is no current record of how breast centres in England deliver biopsy results from breast assessment. Despite policy recommendations, some centres appear to routinely deliver non-cancer results by telephone. Furthermore, it is assumed that cancer results are all delivered in-person as guidelines recommend. There is a current lack of evidence about how often telephone results are used to deliver breast assessment results.[13]

In this study, we aimed to record how breast screening centres in England deliver biopsy results from breast assessment and answer the following questions:

1. How often are telephone results delivered and by whom? Does this differ when results are non-cancer versus cancer?
2. Is there a time difference between results delivered by telephone versus results delivered in-person?

## METHODS
### Participants
A link to an online survey hosted by the Bristol Online Survey platform was sent to all breast screening centres in England on 2 June 2017. At this time, 79 breast screening units existed in England, as confirmed by a list from the Quality Assurance Lead for Breast Screening. Data collection ended on 28 February 2018.

The survey link was distributed to the manager of each breast screening centre via the Quality Assurance lead for each region (East Midlands, East of England, London, North East Yorkshire & Humber, North West, South East, South West, West Midlands). The link was accompanied by a brief study description. The survey was completed by a representative member of staff from each breast screening centre.

Survey completion reminders were sent periodically by the Quality Assurance leads to non-responding centres.

Ethical approval for the survey was obtained from the Biomedical & Scientific Research Ethics Committee at the University of Warwick (REGO-2017–1908).

### Survey piloting and instrument
The survey was designed using a previous tool developed by Clinical Nurse Specialist in Breast Care (Margaret Casey) in combination with discussions with key stakeholders from the NHSBSP. Stakeholders included the programme manager for the NHSBSP, the Quality Assurance Lead for the NHSBSP, and a clinical nurse specialist in breast care. Following this, a draft version underwent three rounds of piloting: stakeholder review of content, stakeholder piloting of the online layout and cognitive interviewing with a layperson.

The main questions in the survey focused on recording how often biopsy results were telephoned. The first question asked about the frequency of delivering benign (non-cancer) biopsy results by telephone (never/occasionally/routinely). The second question asked about the frequency of delivering cancer biopsy results by telephone (never/occasionally/routinely). After these two questions, a free text box was added to allow responders to comment on the answers provided. The survey also recorded who is responsible within the team for delivering telephone results at the centre (clinical nurse specialist, radiologist, radiographer, breast care surgeon, administrative staff, other).

One question recorded the amount of time taken between clinic assessment and the delivery of a result (options spanning between 1 day and >12 days). These data were collected to compare the length of time taken to deliver results for centres who delivered results by telephone routinely versus those who never deliver telephone results.

The survey included nine questions. It was expected that the survey would take 10 min to complete (see the online supplementary appendix 1 for a full survey).

### Data analysis
#### Quantitative
Data cleaning processes were implemented. This involved checking for missing data, coding centres by region and removing duplicate responses. Descriptive statistics and response rates were calculated.

Percentages for delivering non-cancer results and cancer results by telephone were calculated separately, alongside frequencies for who delivers each type of results.

**Table 1** Number of centres that responded to the online survey and provided optional qualitative comments

| Centres in each region (n=79)* | Centres who responded to the survey† (N=63) | Centres who commented on telephoning non-cancer results‡ (N=28) | Centres who commented on telephoning cancer results§ (N=20) |
|---|---|---|---|
| East Midlands (n=9) | 7 | 5 | 3 |
| East of England (n=11) | 10 | 5 | 2 |
| London (n=6) | 4 | 0 | 0 |
| North East Yorkshire and Humber (n=12) | 10 | 3 | 1 |
| North West (n=11) | 8 | 2 | 2 |
| South East (n=8) | 4 | 2 | 3 |
| South West (n=13) | 11 | 5 | 6 |
| West Midlands (n=9) | 9 | 6 | 3 |

*The total number of centres within each region of England.

†The number of centres who completed the survey from each region, providing quantitative data.

‡The number of centres who provided comments in the first qualitative free-text box, asking about telephoning non-cancer results. This was not a mandatory question and not all centres who completed the survey provided qualitative data.

§The number of centres who provided comments in the second qualitative free-text box, asking about telephoning cancer results. This was not a mandatory question and not all centres who completed the survey provided qualitative data.

To identify whether there were any regional trends in the delivery of telephone results, a chi-square was calculated, comparing region (East Midlands, East of England, London, North East Yorkshire & Humber, North West, South East, South West, West Midlands) with telephone frequency (routinely, occasionally and never).

To identify any potential time differences between telephone and in-person results a $\chi^2$ was calculated, comparing telephone frequency (routinely, occasionally and never) with the length of time between assessment and results (1–12+ days).

Statistical significance was assumed at p<0.05. Statistical analysis was conducted using IBM SPSS Statistics 24 software.

### Qualitative

Qualitative free-text comments were analysed using qualitative content analysis.[14 15] This approach allowed for commonalities among staff viewpoints to be identified and to be described narratively, in order to contextualise and expand on the quantitative findings. Intercoder reliability was used to ensure the rigour and trustworthiness of the analysis.[14 16] The analysis was conducted by the lead author (SW) and checked by a second author (DE) to ensure that the meaning of original staff comments was retained. Any disputes in interpretation were resolved by a third author (HS). Qualitative analysis was conducted using NVivo 12.

## RESULTS
### Respondents

Of the 79 breast screening centres in England, 63 (79%) responded to the online survey. All regions were represented in the quantitative survey (table 1).

Data relating to the mean age of women screened at each centre were removed from the data set due to 61.4% of responses being 'I do not know'.

### Quantitative findings
#### Frequency of results by telephone

When delivering non-cancer results, the majority of centres routinely telephoned results (table 2) and most of these results were delivered by clinical nurse specialists (table 3).

When delivering cancer results, the majority of centres never telephoned results (table 2). When cancer results were delivered by telephone, most of these results were delivered by clinical nurse specialists (table 3).

#### Regional differences: frequency of telephone results

No relationship was found between region and the frequency of non-cancer telephone results, ($X^2$ (14, n=63)=11.55, p=0.64). This indicates no regional trends in the use of telephone results services.

#### Time difference: telephone results vs. in-person results

The mean time to deliver results for all centres was 7.03 days (SD=2.03). See table 4 for all means (SDs). No relationship was found between the frequency of telephone results and length of time from assessment to receipt of results, ($X^2$ (16, n=63)=12.76, p=0.69). This indicates that

**Table 2** Frequency and percentage of centres who routinely, occasionally or never deliver results by telephone for non-cancer and cancer results (N=63)

| | Routinely | Occasionally | Never |
|---|---|---|---|
| Non-cancer | 31 (49.2%) | 24 (38.1%) | 8 (12.7%) |
| Cancer | 0 (0%) | 15 (23.8%) | 48 (76.2%) |

**Table 3** Frequency of who delivers results by telephone, for non-cancer and cancer results (N=63)

| | Clinical nurse specialist | Radiologist | Radiographer | Breast surgeon | Administrative staff | Other |
|---|---|---|---|---|---|---|
| Non-cancer | 42 | 17 | 5 | 5 | 0 | 2 |
| Cancer | 13 | 2 | 0 | 2 | 0 | 0 |

N.B. Centres could select multiple responses to this question.

centres delivering results by telephone do not deliver them sooner after the assessment visit than centres delivering results in-person (or never telephone).

### Qualitative findings

In the survey, NHS staff had the option to comment in free-text boxes after two questions. The results are presented in two sections, with one focusing on the first question (non-cancer) and one focusing on the second question (cancer). All regions (excluding London) provided qualitative free-text responses see (table 1). See the online supplementary appendix 2 for a tabular representation of all qualitative comments from the content analysis.

### WHEN DELIVERING BENIGN (NON-CANCER) BIOPSY RESULTS ARE WOMEN NEVER TELEPHONED WITH RESULTS, OCCASIONALLY TELEPHONED WITH RESULTS OR ROUTINELY TELEPHONED WITH RESULTS?

This section presents the qualitative findings following the first survey question (n=28), which asked the frequency of delivering non-cancer biopsy results by telephone (never/occasionally/routinely).

Content analysis revealed that seven centres scheduled for all women to return to receive results in-person. Then, if the test results for these women are confirmed as non-cancer, centres attempt to ring women with telephone results ahead of the prescheduled in-person appointment. Example comments include:

> Women have a scheduled face to face appointment for results but if it's benign we ring them. (Centre ID 02)

> Our aim is to call all the benign results and offer to cancel the booked appointment. (Centre ID 50)

> All women are given a results appointment during assessment clinic. Following MDT, all those with benign biopsy results are contacted by telephone. If contact is made, the result is discussed and the results appointment cancelled. (Centre ID 38)

One centre (Centre ID 25) commented that they 'normally see women face to face', with this being their routine practice. This suggests that breast screening centres differ in how they deliver non-cancer breast assessment results.

### Option to still attend

Content analysis revealed five centres who commented that, when women are contacted by telephone with a non-cancer result, they are still offered the option to attend in-person if they have further questions. Example comments include:

> All patients are given an appointment to attend for results we do telephone with results but patients are still able to attend, and some do. (Centre ID 45)

> After the MDT. Patients are telephoned with benign results by a qualified Breast Care Nurse. They are then offered an OPA with a consultant surgeon if they have concerns. (Centre ID 31)

### Rare and exceptional circumstances

Content analysis revealed five centres only deliver non-cancer results by telephone in exceptional circumstances. Example comments include:

> Only in exceptional circumstances (Centre ID 59)

> This is not done routinely and very rarely occurs. (Centre ID 78)

Reasons given for giving non-cancer results by telephone included if the woman finds it difficult to attend in-person and so the woman can be where she wants to be to receive results.

### Giving women a choice or at patient request

Content analysis revealed five centres ask women how they would like their results to be delivered if it is not cancer. Example comments include:

> We will always offer them an appointment to come in, but the BCNs will ask if they want a telephone call at the time of assessment (Centre ID 63)

> Women are asked at assessment if they would like a telephone call or they can come back for results if they do not wished to be telephoned (Centre ID 29)

**Table 4** Mean (SDs) number of days between assessment and delivery of results for centres who routinely, occasionally and never benign (non-cancer) telephone results

| | Routinely telephone | Occasionally telephone | Never telephone |
|---|---|---|---|
| Mean (SDs) number of days between assessment and delivery of results | 6.77 (1.76) | 7.13 (2.09) | 7.75 (2.81) |

Women are given a choice about how they receive their results when the imaging suggests a benign process (Centre ID 42)

One centre commented that non-cancer results are only delivered by telephone at the request of the patient:

This is not routine practice but happens if a patient requests it and the probability of a benign result is very high. (Centre ID 73)

## SUMMARY OF NON-CANCER CONTENT ANALYSIS

Content analysis revealed conflicting centre comments. Some centres schedule in-person results appointments for all women, but then attempt to contact women with non-cancer results by telephone instead. However, one centre commented that delivering non-cancer results by telephone was not routine practice. Other centres commented that telephoning non-cancer results only happens under exceptional circumstances, such as the woman being unable to attend in-person.

Content analysis revealed that some women who are telephoned with results are still offered the option to attend if they have further questions. Some centres ask women how they would prefer to be contacted with their result if it is not cancer.

## WHEN DELIVERING CANCER BIOPSY RESULTS ARE WOMEN NEVER TELEPHONED WITH RESULTS, OCCASIONALLY TELEPHONED WITH RESULTS OR ROUTINELY TELEPHONED WITH RESULTS?

This section presents the qualitative findings following the second survey question (n=20), which asked the frequency of delivering cancer biopsy results by telephone (never/occasionally/routinely).

Content analysis revealed four centres routinely deliver cancer results in-person. Example comments include:

Cancer diagnoses are always communicated face to face. (Centre ID 39)

[telephone results] This would never be planned. (Centre ID 09)

All positive results or complicated cases are invited back to be given results by the Breast Surgery Team. (Centre ID 26)

### Rare and exceptional circumstances

Content analysis revealed 11 centres only deliver cancer results by telephone in exceptional circumstances. Example comments include:

Rarely telephoned with a cancer diagnosis always at the patients request in extenuating circumstances. (Centre ID 45)

This is a rare occurrence and is only agreed to with the patients prior consent on the understanding they may be receiving a cancer diagnosis. (Centre ID 11).

Very rare - this would only happen with prior agreement if a woman is to be away for an extended period of time. (Centre ID 38)

Reasons for delivering cancer results by telephone under these exceptional circumstances were if the woman finds it difficult to attend in-person and if the woman was going to be away for an extended period of time.

### Giving women a choice or at patient Request

Content analysis revealed two centres ask women how they would like their result to be delivered if it is cancer. Example comments include:

They are asked if the result was a surprise and was a breast cancer would you still wish to get that news over the phone. (Centre ID 01)

Women are asked at assessment if they would like a telephone call or they can come back for results if they do not wished to be telephoned. (Centre ID 29)

Content analysis revealed five centres only deliver cancer results by telephone at the request of the patient. Example comments include:

Patient request only (Centre ID 33)

Only very rarely and at patient's specific request (Centre ID 08)

### Unexpected results and a negative reaction

One centre (Centre ID 60) addressed the issue of how to deal with an unexpected result. At the breast assessment visit, this centre informs women with a high suspicion of a non-cancer result that they will have their result delivered by telephone. However, if 'If there is a positive result which was unexpected a Breast Care Nurse rings the woman to advise an appointment is required to discuss the results'.

One centre (Centre ID 15) commented about a woman who 'reacted extremely badly on telephone' when receiving a cancer result.

## SUMMARY OF CANCER CONTENT ANALYSIS

For cancer results, in-person results are routine. Telephoning cancer results is only being offered under rare or exceptional circumstances such as when women have difficulty in attending. These telephone results are only given at the request of the patient.

Content analysis revealed the potential difficulties in delivering and scheduling results by telephone. One comment highlighted the negative reaction of a woman who received her cancer result by telephone. Another comment addressed the issue of how to deal with a result which was presumed to be non-cancer but turned out to be cancer.

## DISCUSSION

The aim of this research was to record how breast screening centres in England deliver biopsy results from

breast assessment by assessing how often telephone results are delivered and by whom. This research also aimed to see if telephone results delivery differs when assessment results are non-cancer versus cancer. Furthermore, this research aimed to assess if there is a time difference between results delivered by telephone versus results delivered in-person.

Our research suggests that centres routinely delivering results by telephone do not deliver them sooner than centres who deliver results in-person. This contradicts articles citing 'speed' as one of the potential advantages of telephone results.[9 17 18]

Our study found that delivering cancer results by telephone is not a common practice for breast screening centres in England. Telephoning cancer results are only used in exceptional circumstances and only at the request of the patient. For example, telephone results might be used if the woman is physically unable to attend in-person (eg, health issues or away from the country for an extended period). When cancer results are telephoned, most of the results are delivered by Clinical Nurse Specialists. The reason why cancer results are rarely telephoned is probably due to the emotional impact of receiving a cancer diagnosis.[19] The extensive literature on 'breaking bad news' in healthcare places importance on the location where results are received to ensure no disturbances.[20] This may help to explain the one comment in the current study where a centre reported the negative reaction of a woman who received a cancer result by telephone. However, this comment was only made by one centre in the study and may not be representative of the population as a whole so this finding should be interpreted with caution.

Another comment in the study highlighted the issues of how to deal with a result which was presumed to be non-cancer at the assessment stage but turned out to be cancer. At the stage of breast assessment, it is unknown if a woman will receive a cancer or non-cancer result. However, breast assessment tests may indicate a higher chance of a non-cancer result. These women might be informed of the lower likelihood of cancer and are offered the opportunity to receive results by telephone. If the biopsy result then confirms unexpected cancer, women may then be telephoned with a cancer result. This may have implications for anxiety and may be avoided by not offering telephone results. However, this was an issue only highlighted by one centre in the study and does not represent the centre comments as a whole.

Our study found that delivering non-cancer results by telephone is routine practice for roughly half of the breast screening centres in England, with most of the results being delivered by Clinical Nurse Specialists. This appears to contrast with breast screening policy guidelines, which state that telephone results should only be used at the patients request and should not be routinely offered. However, the qualitative findings clarified that some of these centres offer women a choice of how they would prefer to receive their results. This suggests that some centres are still acting within the guidelines by only telephoning women who choose this communication. Offering women the choice between telephone and in-person communication may not be feasible for all centres.[21] For centres who already routinely telephone results, there may be a reduced capacity to provide results in-person if this is requested by the woman. From the content analysis in the current study, a compromise is to telephone all non-cancer results routinely but to also offer the option for the woman to still attend the clinic in-person. Offering patients a choice of a communication method of results at the assessment visit might also be problematic due to heightened anxiety with some women not wanting to make a decision.[22 23]

Some centres who routinely deliver non-cancer results by telephone do not offer patient choice. A common practice is for centres to give all women who attend breast assessment an in-person appointment to return for results, but then telephone women with non-cancer results ahead of this scheduled appointment. The centres commented that this process has the potential to reduce the expected wait time for women to receive results, thus minimising the amount of time spent anxiously waiting.[24] However, the psychological impact of receiving an 'unexpected' communication method has not been considered in this setting. When a telephone result is not expected, it is possible that women may feel unprepared or not in the right mind-set to comprehend the information given.[25] This may contribute to the anxiety associated with screening and might be avoidable harm. However, from the current research, we do not know if this is the case.

## Strengths

This study formally reports the national communication practice of NHSBSP centres for delivering non-cancer and cancer breast assessment results. The high response rate indicated that this is an important issue for staff working within the screening programme.

The content analysis of qualitative comments allowed for the expansion of quantitative survey data, which gave greater insight into how telephone results were implemented in practice.

## Limitations

The time difference between telephone results and in-person results was quantified to allow the survey to be easily answered. This was quantified by the difference in days between clinic assessment and the delivery of a result for centres who either routinely, occasionally or never telephone results. However, other factors could be involved in the speed of results delivery.

The survey responses completed by centre staff were subjective and could not be validated by records. Furthermore, the staff member completing each survey was not recorded as part of the study. Therefore, quantitative data may not accurately reflect current communication practices due to the potential for human error or level of experience of the staff member completing the survey.

It is possible that centres who did not respond to the survey may be systematically different from centres who did. Therefore, the results may not be representative of all breast screening centres in England. However, the high response rate limits this issue.

A formal analysis of the concordance between communication practice and policy guidelines was not conducted. This was considered in the discussion but future research could consider a formal comparison-based analysis.

## CONCLUSIONS

In the NHSBSP, breast assessment results that are cancer are routinely delivered in-person, as recommended by policy guidelines. However, non-cancer breast assessment results are often routinely delivered by telephone despite the recommendations made in policy guidelines. Despite this, telephone results do not appear to be quicker than in-person results in practice.

A common process is to give all women attending breast assessment an appointment to come back to receive results in-person, to then telephone all women with non-cancer results ahead of this appointment. Some centres offer women a choice, although this might not be feasible for all centres and it is possible that women might be too anxious to make an informed decision.

### Research and practice implications

Now that we have a record of current practice, more research is needed in order to fully understand what impact telephone results services have on women attending breast assessment and whether variations in the results giving process also have an impact. This would be particularly beneficial to consider for non-cancer results, where results are being routinely delivered by telephone to large numbers of women every year. Further research should also consider how women themselves might prefer to receive their results and focus on the patient perspective.

**Author affiliations**
[1]Division of Health Sciences, University of Warwick, Coventry, UK
[2]School of Nursing, Midwifery and Health, Coventry University, Coventry, UK
[3]Clinical Trials Unit, University of Warwick, Warwick Medical School, Coventry, UK
[4]NHS Breast Screening Programme, Public Health England, Sheffield, UK
[5]Royal Wolverhampton NHS Trust, Wolverhampton, UK
[6]National Lead Breast Screening QA, Public Health England, Birmingham, UK
[7]Warwick Medical School, University of Warwick, Coventry, UK

**Contributors** SZW: is the lead researcher; collected and analysed the data. SZW, RJ, DRE, ST-P and HKS: drafted the manuscript. SZW, RJ, HKS, ST-P, JJ, MC and OK: participated in the design of the study. OK and JJ: assisted with the dissemination of the survey. All authors have reviewed and approved the manuscript.

**Funding** This research is part of a PhD award and is funded by the Economic and Social Research Council (ESRC) Doctoral Training Centre at the University of Warwick. The funding has been awarded for this studentship to SZW for her PhD project for 4 years of full-time study. The award consists of payment of academic fees and a maintenance award. A further contract between the University of Warwick, Public Health England (PHE) and the PhD Student (SZW) has secured £4000 in research expenses. ST-P is funded by an NIHR Career Development Fellowship. The views expressed in this paper are those of the authors and not the NIHR, PHE, the Department of Health and Social Care, or the Economic and Social Research Council (ESRC).

**Competing interests** None declared.

**Patient consent for publication** Not required.

**Provenance and peer review** Not commissioned; externally peer reviewed.

**Data availability statement** No data are available.

**ORCID iDs**
David R Ellard http://orcid.org/0000-0002-2992-048X
Sian Taylor-Phillips http://orcid.org/0000-0002-1841-4346

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
