## [Reviewer comments · BMJ Open]

ARTICLE DETAILS

TITLE (PROVISIONAL)	Communicating biopsy results from breast screening assessment: Current practice in English breast screening centres and staff perspectives of telephoning results
AUTHORS	Williamson, Sian Zena; Johnson, Rebecca; Sandhu, Harbinder Kaur; Ellard, David R; Jenkins, Jacquie; Casey, Margaret; Kearins, Olive; Taylor-Phillips, Sian

VERSION 1 – REVIEW

REVIEWER	Cornelia J Baines Dalla Lana Scholl of Public Health, University of Toronto, Canada
REVIEW RETURNED	31-Jan-2019

GENERAL COMMENTS	Telephone results services in English breast screening centres.... By Williamson S et al. Reviewed by Cornelia J Baines, January 30, 2019 General Comments: I appreciate the work done by this PhD student and would encourage her not to be dismayed by the comments I have found necessary to make. But I will be forthright. The title could be improved: "Communication practices in English screening centers: how participants learn of their results." Three nouns in a row is not desirable. The abstract: Screening is not diagnosis. Women can be told that their mammogram is suspicious or not suspicious. To say that "screening results are non-cancer or cancer" is not only unacceptably simplistic but inaccurate. Women can be told their mammograms appear within normal limits or they do not. A few will be told that their mammograms are suspicious and of those some will be false positives and others cancer (I won't mention over-diagnosis). Thus the objective as specified in this paper definitely needs revision. And from this problem all the following text is under-mined. The methodology seems appropriate however the results section is not readily understandable – particularly with respect to its last sentence. The conclusion: It is bizarre that the conclusions make no mention of "cancer" results in contrast to what the objective stated (cancer and non-cancer). And although the original title promised communication trends quantifying what centers did, the conclusion says the main finding was anxiety in the participants. Greater clarity is needed and I hope this will be achieved with the help of thesis supervisors. I did read critically all the text but what is written above seems sufficient. I focus on the abstract because it so well reflects the problems in the paper and because so often only an abstract will be read. It also surprised me as I checked the references almost half date from the 90s or the first decade of the 2000s.
--

REVIEWER	Hannah Long, David P. French HL - University of Manchester, UK DPF - University of Manchester, UK
REVIEW RETURNED	25-Feb-2019

GENERAL COMMENTS	Thank you for the opportunity to review this paper. This mixed-methods survey study is on current trends for telephoning results in the NHS BSP, which is an interesting, novel topic. I am very glad to see research is being conducted in this area, for it is much needed. However, there are issues with the paper that need to be addressed, particularly related to the qualitative analysis, aligning the aims and conclusions, and clarifying the type(s) of screening result(s) the paper has investigated. As a result, I have made a number of suggestions for improving the paper; some of these recommendations are quite substantial, but I strongly encourage the authors to persevere because of the originality and importance of this research. Major comments 1. There are issues related to the quality of the qualitative analysis and the quality of its reporting. The narrative is descriptively thin and there is too much (over)interpretation of how women feel in places where you do not appear to have the data to support it. There is a lack of descriptive and/or interpretative narrative specifically related to the NHS staff's views of giving telephone results. In numerous places, the NHS staffs views have been interpreted in terms of how it could make women feel, but (a) this does not address your third research question and (b) you do not have the data to infer how women feel. Together, these issues cause me to wonder whether your survey design elicited sufficiently rich qualitative data for a rigorous thematic analysis, in order to answer your research questions. As a result, I question the suitability of thematic analysis to analyse your qualitative data. I do not feel the paper could be published with the qualitative analysis in its current form. I have a few suggestions that the authors may want to consider. Most importantly, I would advise you to consider how to better represent NHS staff views in order to address your third research question with your qualitative results. It might be worth considering re-analysing your qualitative data using content analysis (see Satu Elo & Helvi Kyngas' 2007 paper for a good overview). This method may be more appropriate because of the type and amount of data you have collected. Alternatively, the authors may want to consider removing the qualitative results from the report in order to really hero the quantitative findings. However, it would be a real shame to lose the qualitative findings altogether, for they could be a real strength of this paper. 2. I would advise you to review the conclusions reported in your Abstract. You have stated that the main finding from this study was the anxiety associated with receiving results by an unexpected communication method. This conclusion is not supported by your results, not least because you have not collected data from women about their feelings of anxiety. Furthermore, in the Abstract and Conclusion, you have suggested that results via telephone may impact women attending screening (i.e. screening uptake), but it is not clear how you have arrived at this conclusion. If this is an implication of your research, you need to make it clearer in your Discussion how your results have led you to suggest this for future research before including it in your Abstract. If these are the issues alluded to by the wording of your Title, I would consider revising
---

	your Title as I am not convinced that you can confidently say these are issues. 3. It is important to clarify how you have defined a non-cancer and cancer result, including where in the screening cycle women received these and standard practice for delivering these results. I think that you have investigated the cancerous or non-cancerous results of screening assessments for recalled women (i.e. the results of further investigations after women have been recalled following an inconclusive screening), but a clearer definition is needed. Doing so will aid understanding of your findings in the wider context of screening. I suggest clarifying this early on in the Background. I recommend making sure it is also clear in the Title and Abstract which result(s) you are investigating. If you are specifically looking at needle biopsy results (this is loosely implied in three places in the Results and Discussion), please include this. If this is the case, it may also be helpful to report in the Background the proportion of women who are recalled and undergo biopsy, so that the reader can get a feel for the volume of telephone and in-person consultations the NHS BSP has to carry out each year. Minor comments 4. Abstracts do not need to be referenced. Please remove the references and re-number your references in-text. 5. The Article Summary asks for the study strengths and limitations, but the final bullet point is neither of these. In the Discussion, you have reported a limitation for the quantitative arm of your study but not the qualitative aspect. I wonder if you could add a limitation of the qualitative arm and then add this to Article Summary in place of the current fourth bullet point. 6. In the Background, how exactly does breast cancer impact over 1.5 million women? Is that the number of diagnoses or deaths from breast cancer, or both? I suggest writing the sentence in your own words (not using it as a quote from the paper) so you can be more specific. 7. In the Background, I think your description of the updated service specification guidelines is a little misleading. My understanding is that the most up to date service specification still advocates that results should be given in person (i.e. in person is still 'gold standard') and that telephone results could be given instead, but only when women specifically ask for them or in cases where there is a strong suspicion that cancer is not present (i.e. only in quite specific circumstances). I can't find any evidence of Public Health England actually recommending results by telephone, as your Background suggests. Please amend this. Relatedly, I think this is what makes your results so interesting – the guidelines advocate delivering results in-person, but many centres are opting for telephone calls. I wonder if you could speculate on why this is the case in your Discussion. Lastly, it would be helpful if you could add another reference or two to the rationale for the debate of delivering results by telephone and/or the concern from Breast Care Nurses – I am interested to learn more but I was not able to access reference number 7 through any of my usual routes (Google Scholar, Google search, my University library, PubMed).
--	---

8. In your Methods, it is interesting that some variables had no/low response and needed to be removed. What was the variable(s) and can you speculate on why the associated question(s) went unanswered by the centres?

9. In your Methods, your response rate from the 77 screening centres is high and this is great to see. However, I was of the impression that there are 79 breast screening units in England (<https://www.england.nhs.uk/wp-content/uploads/2017/04/Gateway-ref-07845-180913-Service-specification-No.-24-NHS-Breast-screening.pdf>). Please could you comment on why the remaining 2 were not invited and change your wording ('all centres in England were invited'), which is misleading. Relatedly, please comment on consenting centres.

10. How did the survey respondents determine the delivery of results? Was it their experienced guess or was it validated by records? If experienced-guess, it may be worth mentioning this as a limitation of the study due to the potential for human error. Or, if through records, this is a strength of your study.

11. Which regions provided qualitative data? The number of centres has been reported; however, as you identify quotes by their region, it would be helpful to know the number of regions that provided qualitative data.

12. The numerous suggestions in this paragraph are related to strengthening the reporting of your thematic analysis and, therefore, whether you wish to take these on board will depend on what you decide to do with your qualitative findings (see comment 1). (a) There is some confusion in your Results regarding the structure of your analysis. You have reported developing 34 codes into 7 categories, which were then refined into themes. Your themes are then written up using sub-headings that do not match the 7 categories. I have deduced from your Appendix that these sub-headings are sub-themes. I advise re-thinking how you have reported the structure of your analysis to address this inconsistency. I'm not sure what the categories really add to your analysis - following Braun and Clarke's thematic analysis method will generally produce codes, sub-themes (sometimes) and themes. Please include the development of sub-themes as a step in your analysis procedure and also expand on the process of thematic analysis, which is currently too brief, e.g. what is involved in coding? Please also report that you have written your narrative up at the sub-theme level. (b) Was your thematic analysis underpinned by a particular theoretical, ontological or epistemological position? Braun and Clarke (and others in the qualitative community) encourage this. If so, please report. If not, please at least report the professional background of those researchers who conducted the analysis in the Methods. (c) In your qualitative results section, you present 10 full length quotes, 4 of which are from South West and 3 are from East Midlands. The majority of your quotes therefore come from a minority of regions. I wonder if you have the data to balance this better. You present more quotes in your Appendix – could any of these be used? (d) In the Discussion, please change your wording in the sentence 'Preparedness was a theme that emerged strongly from the data.' (i) Preparedness is a sub-theme; the theme is 'Managing telephone results: process and preparation'. (ii) Themes do not 'emerge' from the data; themes are actively generated or

	developed by the analysts (Braun and Clarke comment on this in their 2006 paper). Similarly, please make the same two changes for the sentence 'Patient choice was another theme that emerged from the data'. 13. Your data shows that there is no time difference for delivering results by telephone or in-person, but your interpretation of this result as its currently written says there 'might' not be a time difference. Please amend and be specific. 14. In the Discussion, you have stated that offering women the opportunity to attend the clinic, after a telephone call, to further discuss their result makes women feel reassured and valued and this is why they do not attend. However, there could be other reasons that women do not feel the need to attend e.g. like you have reported in your results, the staff believed that many women have personal and logistical issues for not returning to the clinic. 15. You have made some suggestions for future research (i.e. women's views of results communication) and practice implications (i.e. capacity of centres to deliver results in-person) of your findings, but I wonder whether you could do more and/or make these implications clearer, perhaps by rejigging and dedicating a paragraph in your Discussion solely to research and practice implications.
--	--

REVIEWER	Ashley Houston The University of Texas MD Anderson Cancer Center
REVIEW RETURNED	04-Mar-2019

GENERAL COMMENTS	Overall this is a timely and interesting research question examining the use of the telephone to deliver breast cancer screening results. 1. Abstract  - May consider adding detail regarding if both cancer and non-cancer results were analyzed. It appears that both were collected but only non-cancer results were analyzed? - There appears to be a period missing at the end of the objective section. 2. Background:  - Authors may consider editing the Background to focus on what appears to be their research question of delivering breast cancer screening results over the phone. Discussion of non-cancer results and false-positive results delivered over the phone may set up the reader to expect an exploration of these specific topics. But, the authors appear to be investigating delivering all results over the phone. Readers may benefit from some clarification in the Background. - Paragraph 1, Second Sentence: Define NHS. - Paragraph 3, Last Sentence: what types of negative impact might telephone results have on women? Authors may consider elaborating more on the concerns regarding delivering telephone results aside from anxiety. More detail may be helpful for the reader. - Paragraph 4, First Sentence: This detail on false-positive results appears to build a case for the importance of educating women on false positives and how that can be completed over the phone. However, I believe the authors are investigating current practices of delivering breast cancer screening results over the phone. Providing more detail on delivering results over the phone
--

	in other health contexts may provide compelling evidence to the reader for this research question. Describing the concern of anxiety associated with false-positives may be one of many considerations for delivering results in this investigation. 3. Methods:  - Survey Piloting and Instrument: Would it be possible to include a screen shot of the survey tool in the appendix? Or a table listing all questions or domains (if the tool is embargoed for research)? More detail on the data collection instrument may be helpful for the reader. - Data Analysis: How was data collection determined to be complete? - Authors may consider adding more detail to the qualitative analysis section on Page 8. Who coded the interviews? Were they independently coded or did multiple coders code each response? How were coding disputes resolved? How was rigor established? Any indication of saturation? 4. Results:  - Were there any differences observed by the type of provider completing the survey? - Page 15, Setting: Authors may consider moving some of the explanation for why certain themes were observed to the Discussion Section. For example, Lines 23-42 on page 15 appear to be the authors' theory as to why women may feel more comfortable with receiving their results at home. Since this was not stated by the staff but more of an inference from the authors, authors may consider moving it to the discussion or rephrasing this point to be based on the findings from the qualitative data. - Throughout the Results Section: It appears that patients were not interviewed for this investigation. Authors may consider framing the results from the perspective of the clinic staff rather than the perspective of the women receiving the results based on interpretation from the authors. These data collected appear to be the clinic staff's perceptions of their patients' experiences and perspectives and therefore not the actual perspectives of the women receiving results. Authors may consider editing this section to clarify that these are the perspectives of the clinical staff or if they would like to discuss the women's perspectives, they may consider providing this detail in the Discussion (For an example: Preparedness, Page 13, Lines 39-49.) 5. Conclusion:  - Limitations: Any additional limitations to consider? Could there be any differences between those clinics who responded and those that did not (e.g., size? Clinical volume?) 6. References:  - May consider reviewing references for typos (e.g., 18, 19, 21) 7. Overall  - It was a pleasure to review this paper. With revisions, this paper could be an important and meaningful contribution to the literature.
--	---

VERSION 1 – AUTHOR RESPONSE

Reviewer 1: comments and responses

Comment	Response
I appreciate the work done by this PhD student and would encourage her not to be dismayed by the comments I have found necessary to make. But I will be forthright.	Thank you for encouragement and your comments. We found them to be constructive and useful in refining the research.
The title could be improved: "Communication practices in English screening centers: how participants learn of their results." Three nouns in a row is not desirable.	Thank you for your recommendation on the title. Based upon the feedback of the other reviewers, we have revised this to be more reflective of the content of the manuscript. Communicating biopsy results from screening assessment: Current practice in English breast screening centres and staff perspectives of telephoning results
The abstract: Screening is not diagnosis. Women can be told that their mammogram is suspicious or not suspicious. To say that "screening results are non-cancer or cancer" is not only unacceptably simplistic but inaccurate. Women can be told their mammograms appear within normal limits or they do not. A few will be told that their mammograms are suspicious and of those some will be false positives and others cancer (I won't mention over-diagnosis). Thus the objective as specified in this paper definitely needs revision. And from this problem all the following text is under-mined.	We have clarified that this is screening assessment throughout the new abstract. We have also taken this feedback to improve the background section to clarify the process of screening, recall and results. (Background, 1st paragraph, pg. 4) "In the UK, over 2 million women attend breast screening every year, where they have an initial mammogram (3). If an abnormality is found during this screening mammogram, women will be recalled to attend screening assessment for further tests which aim to confirm whether the abnormality found is a cancer" The paper objective has been clarified in the abstract (pg. 2): "To record how breast screening centres in England deliver all biopsy results (cancer/non-cancer) from the screening assessment visit." This has also been clarified in the paper (Background, final paragraph, pg. 5) "In this study, we aimed to record how breast screening centres in England deliver biopsy results from screening

	assessment and answer the following questions:  1) How often are telephone results delivered and by whom? Does this differ when results are non-cancer versus cancer? 2) Is there a time difference between results delivered by telephone versus results delivered in-person?"
The methodology seems appropriate however the results section is not readily understandable – particularly with respect to its last sentence.	The abstract has been rewritten based upon the qualitative data being reanalysed following comments from reviewer 2 (pg. 2). We hope the abstract now has a results section which is more understandable.
The conclusion: It is bizarre that the conclusions make no mention of “cancer” results in contrast to what the objective stated (cancer and non-cancer).	We have added in extra details about the cancer results to the abstract conclusion. “In the NHS Breast Screening Programme, screening assessment results that are cancer are routinely delivered in-person. However, non-cancer screening assessment results are often routinely delivered by telephone, despite policy recommendations.”
And although the original title promised communication trends quantifying what centers did, the conclusion says the main finding was anxiety in the participants. Greater clarity is needed and I hope this will be achieved with the help of thesis supervisors. I did read critically all the text but what is written above seems sufficient. I focus on the abstract because it so well reflects the problems in the paper and because so often only an abstract will be read.	The original title has now been changed to the following, to better reflect the content: Communicating biopsy results from screening assessment: Current practice in English breast screening centres and staff perspectives of telephoning results The conclusion of the paper (see pg. 20) and abstract have also been changed. This is based on the recommendations made here and a recommendation from the second reviewer regarding changing the analysis of the qualitative data. We hope manuscript and abstract conclusion now better reflect what has been found in this research.
It also surprised me as I checked the references almost half date from the 90s or the first decade of the 2000s.	The reference list has now been updated (pg. 22) with some more current evidence. Some of the older references remain as these are large scale reviews which have

	not been updated or are relevant policy documents.
--	--

Reviewer 2: comments and responses

Comment	Response
Thank you for the opportunity to review this paper. This mixed-methods survey study is on current trends for telephoning results in the NHS BSP, which is an interesting, novel topic. I am very glad to see research is being conducted in this area, for it is much needed. However, there are issues with the paper that need to be addressed, particularly related to the qualitative analysis, aligning the aims and conclusions, and clarifying the type(s) of screening result(s) the paper has investigated. As a result, I have made a number of suggestions for improving the paper; some of these recommendations are quite substantial, but I strongly encourage the authors to persevere because of the originality and importance of this research.	Thank you for your kind feedback. We are glad you find the topic to be interesting and of importance to the field. We appreciate your comments which we have found constructive and useful in refining the research.
1. There are issues related to the quality of the qualitative analysis and the quality of its reporting. The narrative is descriptively thin and there is too much (over)interpretation of how women feel in places where you do not appear to have the data to support it. There is a lack of descriptive and/or interpretative narrative specifically related to the NHS staff's views of giving telephone results. In numerous places, the NHS staffs views have been interpreted in terms of how it could make women feel, but (a) this does not address your third research question and (b) you do not have the data to infer how women feel. Together, these issues cause me to wonder whether your survey design elicited sufficiently rich qualitative data for a rigorous thematic analysis, in order to answer your research questions. As a result, I question the suitability of thematic analysis to analyse your qualitative data. I do not feel the paper could be published with the qualitative analysis in its current form. I have a few suggestions that the authors may want to consider. Most importantly, I would advise you to consider how to better represent NHS staff views in	We thank you for your comments regarding the qualitative analysis. On reflection, we agree that the results have been overinterpreted due to being so embedded within the topic. Your fresh view on this has been insightful. We agree that the qualitative findings add value to the report and we were therefore reluctant to remove them entirely. Based upon your suggestion, we have reanalysed the data using qualitative content analysis, focusing purely on staff comments/viewpoints. We have been cautious and practical in our interpretation of the findings, being wary not to speculate too far into the patient viewpoint (which was not the focus of the article). In reanalysing the data, we believe we have now summarised the true comments made by staff and related this back to quantitative findings to give an overall picture of how results are currently delivered in the NHSBSP.

order to address your third research question with your qualitative results. It might be worth considering re-analysing your qualitative data using content analysis (see Satu Elo & Helvi Kyngas' 2007 paper for a good overview). This method may be more appropriate because of the type and amount of data you have collected. Alternatively, the authors may want to consider removing the qualitative results from the report in order to really hero the quantitative findings. However, it would be a real shame to lose the qualitative findings altogether, for they could be a real strength of this paper.	The method is described on pg. 8 in the qualitative paragraph. The rewritten qualitative results section begins on pg. 11.
2. I would advise you to review the conclusions reported in your Abstract. You have stated that the main finding from this study was the anxiety associated with receiving results by an unexpected communication method. This conclusion is not supported by your results, not least because you have not collected data from women about their feelings of anxiety. Furthermore, in the Abstract and Conclusion, you have suggested that results via telephone may impact women attending screening (i.e. screening uptake), but it is not clear how you have arrived at this conclusion. If this is an implication of your research, you need to make it clearer in your Discussion how your results have led you to suggest this for future research before including it in your Abstract. If these are the issues alluded to by the wording of your Title, I would consider revising your Title as I am not convinced that you can confidently say these are issues.	Once again, we agree with the comments made. As a result of these comments (and the reanalysis of the qualitative data) we have updated the abstract to best reflect the conclusions of the manuscript (pg. 2). Following feedback from yourself, and reviewer 1, the title has been revised to the following: Communicating biopsy results from screening assessment: Current practice in English breast screening centres and staff perspectives of telephoning results We feel that this reflects the content of the article more clearly.
3. It is important to clarify how you have defined a non-cancer and cancer result, including where in the screening cycle women received these and standard practice for delivering these results. I think that you have investigated the cancerous or non-cancerous results of screening assessments for recalled women (i.e. the results of further investigations after women have been recalled following an inconclusive screening), but a clearer definition is needed. Doing so will aid understanding of your findings in the wider context of screening.	This has now been clarified in the background (pg. 4, paragraph 1). Breast cancer is one of the most common cancers internationally (1). The National Health Service Breast Screening Programme (NHSBSP) was launched to aid the early detection of breast cancer at the population level, because early detection is linked with better prognosis (2). In the UK, over 2 million women attend breast screening every year,

I suggest clarifying this early on in the Background. I recommend making sure it is also clear in the Title and Abstract which result(s) you are investigating. If you are specifically looking at needle biopsy results (this is loosely implied in three places in the Results and Discussion), please include this. If this is the case, it may also be helpful to report in the Background the proportion of women who are recalled and undergo biopsy, so that the reader can get a feel for the volume of telephone and in-person consultations the NHS BSP has to carry out each year.	where they have an initial mammogram (3). If an abnormality is found during this screening mammogram, women will be recalled to attend screening assessment for further tests which aim to confirm whether the abnormality found is a cancer. These tests can include a core needle biopsy, which involves the removal of sections of tissue from the suspicious breast region which are sent for cytological examination. A biopsy is the definitive test at screening assessment to confirm if the mammogram abnormalities found are cancer. This information has been added to the background (end of first paragraph, pg. 4) “A biopsy is the definitive test at screening assessment to confirm if the mammogram abnormalities found are cancer. In 2014-15, 1,795,307 women in England were screened by the NHSBSP. 68,973 of these women were recalled to attend screening assessment and 31,926 women had a biopsy.”
Abstracts do not need to be referenced. Please remove the references and re-number your references in-text.	References have been removed from the abstract and re-numbered in-text as recommended.
The Article Summary asks for the study strengths and limitations, but the final bullet point is neither of these. In the Discussion, you have reported a limitation for the quantitative arm of your study but not the qualitative aspect. I wonder if you could add a limitation of the qualitative arm and then add this to Article Summary in place of the current fourth bullet point.	The final bullet point has been removed from the article summary. The third bullet point has been revised based on the qualitative reanalysis. “The qualitative comments made by NHS staff gave deeper insight into how telephone results are delivered in practice.” A fourth bullet point has been added with a limitation for the quantitative arm. “Survey responses were subjective and not checked against centre policy documents.”
In the Background, how exactly does breast cancer impact over 1.5 million women? Is that the number of diagnoses or deaths from breast cancer, or both? I suggest writing the sentence in your own words (not using it as a quote from the paper) so you can be more specific.	This sentence has been replaced due to the original reference not giving further detail about the meaning. This has been replaced with the following sentence, supported by a recent reference:

	“Breast cancer is one of the most common cancers internationally.”
In the Background, I think your description of the updated service specification guidelines is a little misleading. My understanding is that the most up to date service specification still advocates that results should be given in person (i.e. in person is still ‘gold standard’) and that telephone results could be given instead, but only when women specifically ask for them or in cases where there is a strong suspicion that cancer is not present (i.e. only in quite specific circumstances). I can’t find any evidence of Public Health England actually recommending results by telephone, as your Background suggests. Please amend this. Lastly, it would be helpful if you could add another reference or two to the rationale for the debate of delivering results by telephone and/or the concern from Breast Care Nurses – I am interested to learn more but I was not able to access reference number 7 through any of my usual routes (Google Scholar, Google search, my University library, PubMed). Relatedly, I think this is what makes your results so interesting – the guidelines advocate delivering results in-person, but many centres are opting for telephone calls. I wonder if you could speculate on why this is the case in your Discussion.	Thanks for your comment – We can see now how this could be misunderstood. The paragraph has now been updated and new references added (background, pg.4, paragraph 2). In recent years, the NHSBSP has been considering which communication method might be preferable for delivering non-cancer biopsy results from screening assessment. The NHSBSP service specification recommends that all screening assessment results should be delivered in-person, which includes cancer and non-cancer results (5). Furthermore, the guidelines state that telephone results should only be used at the patients request and should not be standard practice (6). Despite these recommendations, there is a suggestion that some centres routinely deliver non-cancer screening assessment results by telephone (5, 7). There is ongoing concern about the impact of delivering a non-cancer assessment result by telephone on screened women. One of the main concerns is how anxious women feel when receiving a result from screening, even when the result is non-cancer (8). Another concern is the potential for miscommunication by telephone. This has been considered in the discussion section (discussion, pg. 17, paragraph 5): “Overall, delivering non-cancer results by telephone is routine practice for roughly half of the breast screening centres in England, with most of the results being delivered by Clinical Nurse Specialists. This contrasts with recommendations in breast screening policy guidelines, which state that telephone results should only be used at the patients request

	and should not be routinely offered. Staff commented that telephone results are used when patients find it difficult to attend in-person. A potential reason that telephone results services are being used against policy recommendations is that telephone results have the potential to be time and cost-efficient. Another potential reason is that women who attend screening may prefer to be contacted by telephone when the result is not cancer.”
In your Methods, it is interesting that some variables had no/low response and needed to be removed. What was the variable(s) and can you speculate on why the associated question(s) went unanswered by the centres?	Added additional detail about the missing variable of average screening age (pg. 9, results) Data relating to the average (mean) age of women screened at each centre were removed from the data set due to 61.4% of responses being ‘I don’t know’.
In your Methods, your response rate from the 77 screening centres is high and this is great to see. However, I was of the impression that there are 79 breast screening units in England. Please could you comment on why the remaining 2 were not invited and change your wording (‘all centres in England were invited’), which is misleading. Relatedly, please comment on consenting centres.	At the time of the research being conducted there were 79 breast screening units and all were invited. This was confirmed by an official list of units from the Quality Assurance Lead for Breast Screening. We have corrected this and clarified where this information came from in the text. (Methods, participants, pg. 6) “A link to an online survey hosted by the Bristol Online Survey platform was sent to all breast screening centres in England on 2nd June 2017. At this time, 79 breast screening units existed in England, as confirmed by a list from the Quality Assurance Lead for Breast Screening. Data collection ended on 28th February 2018.”
How did the survey respondents determine the delivery of results? Was it their experienced guess or was it validated by records? If experienced-guess, it may be worth mentioning this as a limitation of the study due	This has been added as a limitation for the study (pg. 19).

to the potential for human error. Or, if through records, this is a strength of your study.	The survey responses completed by centre staff were subjective and could not be validated by records. Furthermore, the staff member completing each survey was not recorded as part of the study. Therefore, quantitative data may not accurately reflect current communication practice due to the potential for human error or level of experience of the staff member completing the survey.
Which regions provided qualitative data? The number of centres has been reported; however, as you identify quotes by their region, it would be helpful to know the number of regions that provided qualitative data.	A sentence has been added about the regions who provided qualitative data (pg. 11, end of first paragraph) and a table has also been provided (pg. 9, Results, Respondents). “All regions (excluding London) provided qualitative free-text responses.”
The numerous suggestions in this paragraph are related to strengthening the reporting of your thematic analysis and, therefore, whether you wish to take these on board will depend on what you decide to do with your qualitative findings (see comment 1). (a) There is some confusion in your Results regarding the structure of your analysis. You have reported developing 34 codes into 7 categories, which were then refined into themes. Your themes are then written up using sub-headings that do not match the 7 categories. I have deduced from your Appendix that these sub-headings are sub-themes. I advise re-thinking how you have reported the structure of your analysis to address this inconsistency. I’m not sure what the categories really add to your analysis - following Braun and Clarke’s thematic analysis method will generally produce codes, sub-themes (sometimes) and themes. Please include the development of sub-themes as a step in your analysis procedure and also expand on the process of thematic analysis, which is currently too brief, e.g. what is involved in coding? Please also report that you	This comment is no longer applicable due to the reanalysis of the data using qualitative content analysis, as recommended in comment 1.

have written your narrative up at the sub-theme level.

(b) Was your thematic analysis underpinned by a particular theoretical, ontological or epistemological position? Braun and Clarke (and others in the qualitative community) encourage this. If so, please report. If not, please at least report the professional background of those researchers who conducted the analysis in the Methods.

(c) In your qualitative results section, you present 10 full length quotes, 4 of which are from South West and 3 are from East Midlands. The majority of your quotes therefore come from a minority of regions. I wonder if you have the data to balance this better. You present more quotes in your Appendix – could any of these be used?

(d) In the Discussion, please change your wording in the sentence 'Preparedness was a theme that emerged strongly from the data.' (i) Preparedness is a sub-theme; the theme is 'Managing telephone results: process and preparation'. (ii) Themes do not 'emerge' from the data; themes are actively generated or developed by the analysts (Braun and Clarke comment on this in their 2006 paper). Similarly, please make the same two changes for the sentence 'Patient choice was another theme that emerged from the data'.

This comment is no longer applicable due to the reanalysis of the data using qualitative content analysis, as recommended in comment 1.

We have, however, added further detail on the researchers who conducted the analysis (see qualitative section, pg. 8).

This comment is no longer applicable due to the reanalysis of the data using qualitative content analysis, as recommended in comment 1.

In the reanalysis, we tried to use a selection of comments from all regions.

	This comment is no longer applicable due to the reanalysis of the data using qualitative content analysis, as recommended in comment 1. Due to the reanalysis, the discussion has been rewritten (see pg. 17). We have avoided discussing themes 'emerging'.
Your data shows that there is no time difference for delivering results by telephone or in-person, but your interpretation of this result as its currently written says there 'might' not be a time difference. Please amend and be specific.	This has been amended to the following (see pg. 10, final paragraph): “This indicates that centres delivering results by telephone do not deliver them sooner after the assessment visit.”
In the Discussion, you have stated that offering women the opportunity to attend the clinic, after a telephone call, to further discuss their result makes women feel reassured and valued and this is why they do not attend. However, there could be other reasons that women do not feel the need to attend e.g. like you have reported in your results, the staff believed that many women have personal and logistical issues for not returning to the clinic.	Due to the reanalysis of the data, the discussion has been rewritten (pg. 17 onwards). As such, we have removed this section.
You have made some suggestions for future research (i.e. women's views of results communication) and practice implications (i.e. capacity of centres to deliver results in-person) of your findings, but I wonder whether you could do more and/or make these implications clearer, perhaps by rejigging and dedicating a paragraph in your Discussion solely to research and practice implications.	We have dedicated a paragraph to research and practice implications at the end of our discussion section (pg. 20).

Reviewer 3: comments and responses

Comment	Response
Abstract May consider adding detail regarding if both cancer and non-cancer results were analyzed. It appears that both were collected but only non-cancer results were analyzed?	Details of the quantitative analysis of cancer results have been added to the abstract. (pg. 2) When delivering cancer results, 76.2% of centres never telephone results and 23.8% of

	centres occasionally telephone results. No centres reported delivering cancer results routinely by telephone. Qualitative comments suggest that cancer results are only telephoned at patient request and under exceptional circumstances.
Abstract There appears to be a period missing at the end of the objective section.	Due a substantial redrafting of the article based on feedback from reviewer 1 and reviewer 2, we believe this period issue has now been resolved.
Background Authors may consider editing the Background to focus on what appears to be their research question of delivering breast cancer screening results over the phone. Discussion of non-cancer results and false-positive results delivered over the phone may set up the reader to expect an exploration of these specific topics. But, the authors appear to be investigating delivering all results over the phone. Readers may benefit from some clarification in the Background	We agree that we focused too much on non-cancer results in the original background. This has now been rewritten to include a more general section on the communication methods used in healthcare (see end of pg. 4). We have also added a clearer section at the end of the background (just before the research questions, pg. 5) to highlight why cancer and non-cancer results were looked at. “There is no current record of how breast screening centres in England deliver biopsy results from screening assessment. Despite policy recommendations, some centres appear to routinely deliver non-cancer results by telephone. Furthermore, it is assumed that cancer results are all delivered in-person as guidelines recommend. There is a currently lack of evidence about how often telephone result are used to deliver screening assessment results.”
Background Paragraph 1, Second Sentence: Define NHS.	This has now been defined (pg. 4).
Background Paragraph 3, Last Sentence: what types of negative impact might telephone results have on women? Authors may consider elaborating more on the concerns regarding delivering telephone results aside from anxiety. More detail may be helpful for the reader.	This section has been rewritten to include the potential for miscommunication (pg. 4, second paragraph). “One of the main concerns is how anxious women feel when receiving a result from screening, even when the result is non-cancer.

	Another concern is the potential for miscommunication by telephone.” We then discuss the use of different methods of communication in other healthcare contexts.
Background Paragraph 4, First Sentence: This detail on false-positive results appears to build a case for the importance of educating women on false positives and how that can be completed over the phone. However, I believe the authors are investigating current practices of delivering breast cancer screening results over the phone. Providing more detail on delivering results over the phone in other health contexts may provide compelling evidence to the reader for this research question. Describing the concern of anxiety associated with false-positives may be one of many considerations for delivering results in this investigation.	We have rewritten this paragraph to reflect the different methods of communication used in other healthcare contexts. (final paragraph. Pg. 4) “In other areas of healthcare, a variety of communication methods are used to deliver results, including face-to-face consultations, telephone, letters and email. Each method of communication used in results delivery presents different advantages, disadvantages and challenges in implementation. Results delivered in-person are often seen as the ‘gold standard’. However, as technology advances, fewer healthcare results are now delivered in-person. Liederman et al. stated that ‘face-to-face contact is not necessary for effective communication’ (pg. 52). Most commonly, results delivery is moving towards telephone and telemedicine. Despite this, Cochrane review evidence suggests that we still do not know enough about the impact of telephone results services on healthcare outcomes. This is concerning due to the rapid changes in communication that may not have been adequately evaluated.”
Methods Survey Piloting and Instrument: Would it be possible to include a screen shot of the survey tool in the appendix? Or a table listing all questions or domains (if the tool is embargoed for research)? More detail on the data collection instrument may be helpful for the reader.	The survey has been included as an appendix.
Methods	This sentence was removed, as the closing date of the online survey (28th February 2018) was

Data Analysis: How was data collection determined to be complete?	mentioned earlier in the participants section, which was when data were considered complete.
Methods Authors may consider adding more detail to the qualitative analysis section on Page 8. Who coded the interviews? Were they independently coded or did multiple coders code each response? How were coding disputes resolved? How was rigor established? Any indication of saturation?	The data has now been reanalysed based on the comments from reviewer 2. However, we have added further details to the methods section (see qualitative section, pg. 8) explaining who coded the data, who checked the data and how disputes were resolved. “The analysis was conducted by the lead author (SW) and checked by a second author (DE) to ensure that the meaning of original staff comments was retained. Any disputes in interpretation were resolved by a third author (HS).”
Results Were there any differences observed by the type of provider completing the survey?	Unfortunately we did not collect the data regarding who completed the survey. We agree that this would have been useful information and appreciate this as a limitation of the study which is included in the discussion (pg. 19). The survey responses completed by centre staff were subjective and could not be validated by records. Furthermore, the staff member completing each survey was not recorded as part of the study. Therefore, quantitative data may not accurately reflect current communication practice due to the potential for human error or level of experience of the staff member completing the survey.
Results Page 15, Setting: Authors may consider moving some of the explanation for why certain themes were observed to the Discussion Section. For example, Lines 23-42 on page 15 appear to be the authors’ theory as to why women may feel more comfortable with receiving their results at home. Since this was not stated by the staff but more of an inference from the authors, authors may consider moving it to the discussion or rephrasing this point to be based on the findings from the qualitative data.	We have moved some of the results details into the discussion and framed them as author thoughts as opposed to direct comments from staff (see discussion section on pg. 17). This has been based on the reanalysis of the data using qualitative content analysis, as recommended by reviewer 2.

Results Throughout the Results Section: It appears that patients were not interviewed for this investigation. Authors may consider framing the results from the perspective of the clinic staff rather than the perspective of the women receiving the results based on interpretation from the authors. These data collected appear to be the clinic staff's perceptions of their patients' experiences and perspectives and therefore not the actual perspectives of the women receiving results. Authors may consider editing this section to clarify that these are the perspectives of the clinical staff or if they would like to discuss the women's perspectives, they may consider providing this detail in the Discussion (For an example: Preparedness, Page 13, Lines 39-49.)	We agree that the comments have been removed from the staff viewpoint context and too much inference was made about the patients views in the original manuscript. We have reanalysed the data based on comments from reviewer 2. We have also modified the article throughout to clarify our focus on staff viewpoints only. Any implications on patients are in the discussion and interpreted more cautiously.
Conclusion Limitations: Any additional limitations to consider? Could there be any differences between those clinics who responded and those that did not (e.g., size? Clinical volume?)	We have added in the limitation as suggested (pg. 19): It is possible that centres who did not respond to the survey may be systematically different from centres who did. Therefore, the results may not be representative of all breast screening centres in England. However, the high response rate limits this issue.
References May consider reviewing references for typos (e.g., 18, 19, 21)	The reference list has been updated and reviewed for typos.
Overall It was a pleasure to review this paper. With revisions, this paper could be an important and meaningful contribution to the literature.	We thank you for your comments and are glad you enjoyed reviewing the paper.

VERSION 2 – REVIEW

REVIEWER	Cornelia J Baines Professor Emerita Dalla School of Public Health University of Toronto Canada
REVIEW RETURNED	24-May-2019

GENERAL COMMENTS	This is the second time I have reviewed this paper and it is clearly much improved. I have only three major comments to make at this time. Table 1 would benefit from some clarification. As I understand it, it might be preferable to enter in brackets the total number of centers in each region in column 1 and not in column 2. It is ambiguous for the header to say number of centers responding and what I interpret as the numbers responding in column 2. Furthermore it is puzzling that if there are four centers responding in London, columns 4 and 5 indicate no responses. This ambiguity could easily be remedied with appropriate footnotes.. Secondly I would prefer to see the qualitative and quantitative findings displayed in tabular form and finally I would suggest that the article would benefit from being more succinct. Telephone results services in English breast screening centres.... By Williamson S et al. Reviewed by Cornelia J Baines, January 30, 2019 General Comments: I appreciate the work done by this PhD student and would encourage her not to be dismayed by the comments I have found necessary to make. But I will be forthright. The title could be improved: Communication practices in English screening centers: how participants learn of their results. Three nouns in a row is not desirable. The abstract: Screening is not diagnosis. Women can be told that their mammogram is suspicious or not suspicious. To say that “screening results are non-cancer or cancer” is not only unacceptably simplistic but inaccurate. Women can be told their mammograms appear within normal limits or they do not. A few will be told that their mammograms are suspicious and of those some will be false positives and others cancer (I won’t mention over-diagnosis). Thus the objective as specified in this paper definitely needs revision. And from this problem all the following text is undermined. The methodology seems appropriate however the results section is not readily understandable – particularly with respect to its last sentence. The conclusion: It is bizarre that the conclusions make no mention of “cancer” results in contrast to what the objective stated (cancer and non-cancer). And although the original title promised communication trends quantifying what centers did, the conclusion says the main finding was anxiety in the participants. Greater clarity is needed and I hope this will be achieved with the help of thesis supervisors. I did read critically all the text but what is written above seems sufficient. I focus on the abstract because it so well reflects the problems in the paper and because so often only an abstract will be read. It also surprised me as I checked the references almost half date from the 90s or the first decade of the 2000s.
---

REVIEWER	Ms Hannah Long & Prof David French University of Manchester. UK
REVIEW RETURNED	02-Aug-2019

GENERAL COMMENTS	We would like to commend the authors on how they have addressed our review comments. There are some minor points suggested below that would enhance the paper further if they were addressed. Abstract – Results
--

	A little clarification would aid the readers understanding of the result in lines 26-27 – telephone results were no faster than what? Results – Quantitative Do the authors have descriptive statistics (mean and SD) for time differences in delivering results? This information would be interesting, and would aid the readers understanding of the Chi square test results and the conclusion that telephone results are not delivered significantly faster than in person results. The sentence on page 11 lines 41-44 could be edited to make it clearer that, as indicated by the quotes, you are referring to face to face results appointments (not to be confused with appointments for screening assessment). Results – Qualitative ‘Thematic analysis’ is reported in the header for the qualitative findings. There are slight inconsistencies between the number of centres that provided qualitative data in response to question 1 in Table 1 (n=28) and in-text in the Results (n=29) and also for question 2 in Table 1 (n=20) and in-text in the Results (n=21). Discussion Only one centre reported dealing with unexpected cancer results and only one centre reported a negative reaction when a woman was given her cancer result over the telephone. The authors have discussed these findings later on the Discussion. As these scenarios were both reported by only one centre, it may be worth mentioning that these are not necessarily representative, i.e. make the discussion more tentative on these points. The authors have stated that the quantitative results show that roughly half of centres routinely deliver non-cancer results by telephone and that this goes against policy guidelines. However, the new qualitative results suggest that a number of centres are delivering non cancer results over the phone because they have given women the choice and this is what women have opted for. This suggests that centres may still be acting within guidelines (which specify that delivering results over the phone is okay if requested by women). The Discussion paragraph (page 17 lines 45 onwards) could be slightly rephrased in places to mention this nuance, as it’s one way in which your qualitative findings have provided deeper insight.
--	--

REVIEWER	Ashley Houston The University of Texas MD Anderson Cancer Center
REVIEW RETURNED	01-Jun-2019

GENERAL COMMENTS	Thank you for the thoughtful edits. It was a pleasure reading this improved version. Minor suggestions below.  1. Abstract  - Authors may consider using qualitative analysis/content analysis rather than qualitative comments. It appears that the qualitative content analysis revealed these different processes.
--

	 - May consider defining NHS in first sentence of Conclusions. - Authors mention policy recommendations but provide no description. Authors may consider removing this or providing a brief explanation of the policy and who recommends this policy to help give context to the reader. 2. Article Summary—Strengths and limitations  - Abbreviations appear not to be defined and may be used inconsistently (NHSBSP and NHS Breast Cancer Screening Programme). - Consider consistent use of periods. - Is there a thought that practice is going against the center policy? I believe that subjective responses should capture the current practice of delivering results, one of the aims of the paper. If an aim is also to assess concordance between practice and center policy, that may be important to analyze. But, if not, authors may reconsider introducing this limitation in this section of the paper because it may be more confusing for the reader. 3. Background:  - Thank you for your edits. 4. Methods:  - Brief description of the qualitative analysis. May consider adding a bit more detail about the checking procedures to help the reader understand the rigor of the analysis. 5. Results:  - Thank you for your edits. 6. Discussion:  - Authors may reconsider use of informal language (“breaking bad news” or “on the other hand”). - Authors may consider describing the qualitative analysis or content analysis of staff comments. When described as “staff comments,” findings do not appear to reflect the content analysis completed. Authors may consider, “Content analysis BMJOpen_2018_028683.R1—Response to Reviewers revealed...” - Second paragraph:  o Authors may consider adding citation for the articles citing “speed.” 7. References:  - Thank you for your edits. 8. Overall  - It was a pleasure to review this paper. With minor revisions, this paper will be an important and meaningful contribution to the literature.
--	--

VERSION 2 – AUTHOR RESPONSE

Reviewer 1: comments and responses

Comment	Response
This is the second time I have reviewed this paper and it is clearly much improved. I have only three major comments to make at this time.	Thank you for your second review of the paper. Your comments have been useful in improving the quality of the manuscript.
Table 1 would benefit from some clarification. As I understand it, it might be preferable to enter in brackets the total number of centers in each region in column 1 and not in column 2. It is ambiguous for the header to say number of centers responding and what I interpret as the numbers responding in column 2. Furthermore it is puzzling that if there are four centers responding in London, columns 4 and 5 indicate no responses. This ambiguity could easily be remedied with appropriate footnotes..	Thank you for your comment. I have amended the table as suggested, by moving the number of centres in each region to the first column. I have also added clarification for the columns via footnotes. Hopefully this clarifies the number of centres who completed the survey and provided quantitative data, followed by the number of those centres who then provided qualitative comments, which were not mandatory.
Secondly I would prefer to see the qualitative and quantitative findings displayed in tabular form	The findings have now been presented in tabular form. The quantitative data is included in Tables 2, 3 & 4. The qualitative data is included in Appendix 2 as we felt that we needed to retain the qualitative in-text comments to illustrate the findings of the content analysis.
Finally I would suggest that the article would benefit from being more succinct.	We hope that providing the data in tabular form has made the manuscript more succinct. We have also gone through the manuscript and condensed some sections where appropriate.

Reviewer 2: comments and responses

Comment	Response
We would like to commend the authors on how they have addressed our review comments. There are some minor points suggested below that would enhance the paper further if they were addressed.	Thank you for your kind words. Also, thank you for taking the time to review this manuscript for the second time. Your comments were helpful in improving the quality and reporting of the research.
Abstract – Results A little clarification would aid the readers understanding of the result in lines 26-27 – telephone results were no faster than what?	Thank you for comment – we have added clarification. “Centres who telephoned results routinely did not deliver results sooner than centres who deliver results in-person.”
Results – Quantitative Do the authors have descriptive statistics (mean and SD) for time differences in delivering results? This information would be interesting, and would aid the readers understanding of the Chi square test results and the conclusion that telephone results are not delivered significantly faster than in person results.	Added in the mean length of time to deliver results and the standard deviation. Added a table for the mean and SD of each group. (Table 4)
The sentence on page 11 lines 41-44 could be edited to make it clearer that, as indicated by the quotes, you are referring to face to face results appointments (not to be confused with appointments for screening assessment).	This has been reworded and is hopefully clearer.
Results – Qualitative ‘Thematic analysis’ is reported in the header for the qualitative findings.	Thanks for spotting this – I have now removed this from the manuscript.
There are slight inconsistencies between the number of centres that provided qualitative data in response to question 1 in Table 1 (n=28) and in-text in the Results (n=29) and also for question 2	This was an error and has been edited to resolve inconsistencies.

in Table 1 (n=20) and in-text in the Results (n=21).	
Discussion Only one centre reported dealing with unexpected cancer results and only one centre reported a negative reaction when a woman was given her cancer result over the telephone. The authors have discussed these findings later on the Discussion. As these scenarios were both reported by only one centre, it may be worth mentioning that these are not necessarily representative, i.e. make the discussion more tentative on these points.	Thank you for your comment. We felt that it was important to keep this comment in, as it adds to the overall picture of results communication in the NHSBSP. We have added in a sentence to clarify that this was one comment and may not be representative, in order to allow readers to interpret this finding with caution. “However, this comment was only made by one centre in the study and may not be representative of the population as a whole so this finding should be interpreted with caution.”
The authors have stated that the quantitative results show that roughly half of centres routinely deliver non-cancer results by telephone and that this goes against policy guidelines. However, the new qualitative results suggest that a number of centres are delivering non cancer results over the phone because they have given women the choice and this is what women have opted for. This suggests that centres may still be acting within guidelines (which specify that delivering results over the phone is okay if requested by women). The Discussion paragraph (page 17 lines 45 onwards) could be slightly rephrased in places to mention this nuance, as it’s one way in which your qualitative findings have provided deeper insight.	The discussion has been amended to reflect the comments made.

Reviewer 3: comments and responses

Comment	Response
Thank you for the thoughtful edits. It was a pleasure reading this improved version. Minor suggestions below.	Thank you for your comments and review of the manuscript. Your comments have been useful in improving the quality and reporting of our research.
1. Abstract - Authors may consider using qualitative analysis/content analysis rather than qualitative comments. It appears that the qualitative content analysis revealed these different processes.	Thanks – I have amended the abstract to say 'qualitative content analysis' for clarity.
May consider defining NHS in first sentence of Conclusions.	The abbreviation has now been defined.
Authors mention policy recommendations but provide no description. Authors may consider removing this or providing a brief explanation of the policy and who recommends this policy to help give context to the reader	We have clarified that these are breast screening policy recommendations. However, we feel as though a full description of the policy would not be possible in the abstract due to the word count restrictions.
2. Article Summary—Strengths and limitations - Abbreviations appear not to be defined and may be used inconsistently (NHSBSP and NHS Breast Cancer Screening Programme).	The abbreviations have now been defined and checked throughout the manuscript for consistency.
Consider consistent use of periods.	Period consistency has been checked and amended in article summary.
- Is there a thought that practice is going against the center policy? I believe that subjective responses should capture the current practice of delivering results, one of the aims of the paper. If an aim is also to assess concordance between practice and center policy, that may be important to analyze. But, if not, authors may reconsider introducing this limitation in this section of the paper because it may be more confusing for the reader.	This has been added as a limitation in the article summary and in the limitations section following the discussion.
3. Background: - Thank you for your edits.	Thank you for your comments.

4. Methods: - Brief description of the qualitative analysis. May consider adding a bit more detail about the checking procedures to help the reader understand the rigor of the analysis.	I have added some references and a sentence describing the inter-coder reliability process to ensure rigour and trustworthiness.
5. Results: - Thank you for your edits.	Thank you for your comments.
6. Discussion: - Authors may reconsider use of informal language (“breaking bad news” or “on the other hand”).	“Breaking bad news” is the terminology used in the literature regarding cancer diagnoses. Therefore, we have retained this phrase in order to remain consistent with the existing evidence. However, we have put this in inverted commas to clarify this. “On the other hand” has been removed.
Authors may consider describing the qualitative analysis or content analysis of staff comments. When described as “staff comments,” findings do not appear to reflect the content analysis completed. Authors may consider, “Content analysis revealed...”	This has been amended throughout the results and discussion to reflect the findings of the qualitative content analysis.
Second paragraph: o Authors may consider adding citation for the articles citing “speed.”	References have been added.
7. References: - Thank you for your edits.	Thank you for your comments.
8. Overall - It was a pleasure to review this paper. With minor revisions, this paper will be an important and meaningful contribution to the literature.	Thank you for your feedback. We are glad you enjoyed reviewing this paper and that you think it will be a meaningful contribution.